# The masked seducers: Lek courtship behavior in the wrinkle-faced bat *Centurio senex* (Phyllostomidae)

Bernal Rodríguez-Herrera[1]*, Ricardo Sánchez-Calderón[1], Victor Madrigal-Elizondo[1], Paulina Rodríguez[1], Jairo Villalobos[2], Esteban Hernández[2], Daniel Zamora-Mejías[1,3], Gloria Gessinger[4,5], Marco Tschapka[4,5]

1 Universidad de Costa Rica, San José, Costa Rica, 2 Estación de Investigación Miguel Alfaro, Hotel Villablanca, San Ramón, Costa Rica, 3 Universidad Nacional Autónoma de México, Mexico DF, Mexico, 4 University of Ulm, Ulm, Germany, 5 Smithsonian Tropical Research Institute, Ancon, Panama

* bernal.rodriguez@ucr.ac.cr

**Data Availability Statement:** All relevant data are within the manuscript and its Supporting Information files.

## Abstract

*Centurio senex* is an iconic bat characterized by a facial morphology deviating far from all other New World Leaf Nosed Bats (Phyllostomidae). The species has a bizarrely wrinkled face and lacks the characteristic nose leaf. Throughout its distribution from Mexico to Northern South America the species is most of the time rarely captured and only scarce information on its behavior and natural history is available. *Centurio senex* is frugivorous and one of the few bats documented to consume also hard seeds. Interestingly, the species shows a distinct sexual dimorphism: Adult males have more pronounced facial wrinkles than females and a fold of skin under the chin that can be raised in style of a face mask. We report the first observations on echolocation and mating behavior of *Centurio senex*, including synchronized audio and video recordings from an aggregation of males in Costa Rica. Over a period of 6 weeks we located a total of 53 perches, where during the first half of the night males were hanging with raised facial masks at a mean height of 2.35 m. Most of the time, the males moved just their wing tips, and spontaneously vocalized in the ultrasound range. Approaches of other individuals resulted in the perching male beating its wings and emitting a very loud, low frequency whistling call. Following such an encounter we recorded a copulation event. The observed aggregation of adult *C. senex* males is consistent with lek courtship, a behavior described from only few other bat species.

## Introduction

Mating systems of bats remain a rather poorly understood topic. A review of McCracken and Wilkinson [1] presented information for only 6.9% of the bat species described at that time (963 spp). This percentage probably has not increased at the same rate as the number of described bats, which currently is at 1406 species (Simmons, pers. comm. 2019). One of the reasons for the general scarceness of information is that, in addition to their nocturnal habits, bats are relatively small and highly mobile, and only over the last decades advanced technology for the nocturnal observation of bats, such as infrared cameras and ultrasound recorders,

**Funding:** The owner of the study site, Hotel Villa Blanca, provided support in the form of salaries for authors J.V. and E.H., but did not have any role in the study design, data collection and analysis, decision to publish, or preparation of the manuscript. The specific roles of these authors are defined in the 'author contributions' section. The authors also want to acknowledge the long-standing cooperation between UCR and UULM, which is supported by the German Academic Exchange Service (DAAD). GG was funded by the Heinrich Böll Foundation. MT acknowledges the financial support of the University of Ulm.

**Competing interests:** The authors have declared that no competing interests exist. The owner of the study site, Hotel Villa Blanca, provided support in the form of salaries for authors J.V. and E.H. This does not alter our adherence to PLOS ONE policies on sharing data and materials.

became more readily available. Additionally, variables such as time and place where bats will show courtship behavior are for many species extremely difficult or impossible to predict and to access. Mainly for this reason, most of the information on bat reproduction available is based on mist net captures, when individuals in hand can be examined for signs of pregnancy and lactation of females and testicle state in males [2]. This may yield information on sex ratio, on the reproductive cycles, however, these morphological diagnoses yield no information on the actual courtship and mating behavior.

Nevertheless, the speciose order Chiroptera is known to have a high variety of mating systems. Observations indicate the existence of promiscuous (*Myotis lucifugus*, [3]) and monogamous (*Vampyrum spectrum*, [4]) mating systems; there are examples of polygyny with defense of females (*Phyllostomus hastatus*, [5]; *Pipistrellus kuhli*, [6]) as well as of polygyny with defense of resources (*Artibeus jamaicensis*, [7, 8], *Desmodus rotundus*, [9]), and there are also rare reports on the existence of lek mating in bats (*Hypsignathus monstrosus*, [10]).

Among all mating systems known from bats, "lek" is the rarest recorded. Leks are clusters of sexually displaying territorial males at a common display ground, where female mate choice [11] causes intense competition among males, thus potentially reducing reproductive chances for individual males. Males offering a better display have in general a higher chance of being chosen by a female [11, 12]. Lek courtship behavior has been first described in birds [13]. The classic lek definition includes at least five ecological and physiological prerequisites [10, 11, 14]: 1) existence of a mating arena, 2) male territories contain no resources (food, water, protection) other than access to males, 3) females have the opportunity to select a male for mating, 4) absence of male parental care, and finally, 5) internal fertilization. There are two categories of leks, the "classic" lek, where males display in close proximity to each other, and "exploded" leks, where males are distributed over a larger area [14]. Lekking behavior includes the transfer of information between the sexes, e.g., in form of visual signals or through vocalizations. Independently of lek systems, such social communication is documented for many bat species and may include visual, tactile, olfactory and acoustic signals. In a few species, males show even sophisticated songs during courtship [15, 16].

Lekking behavior occurs in a wide range of vertebrates and invertebrates but is not common. There are well-documented lek systems in insects, such as in drosophilid and tephritid fruit flies (for a complete list see [17]). In vertebrates, lek behavior is mostly reported from birds, where it has been documented from at least 148 species [18], while occasionally the behavior occurs also in amphibians (e.g., [19]), fish (e.g., [20]) and reptiles (e.g., [21]). In mammals, classic lek behavior appears to be rare and is found only in 12 species, mainly ungulates and pinnipeds, while 14 other species show a lek-like behavior that not fully matches the criteria mentioned above [22].

*Centurio senex* GRAY 1842 is an extraordinary phyllostomid species distributed from Mexico throughout Central America to Venezuela and Trinidad and Tobago [23]. Throughout this wide range, it is in general only rarely captured, which accounts for the scarce information available. *Centurio senex* is predominantly frugivorous, but is also one of the few neotropical bats documented to consume hard seeds [24, 25]. The species shows a high degree of–still unexplained–facial modifications that deviate far from the general morphology of New World Leaf Nosed Bats (Phyllostomidae). *Centurio senex* has a uniquely wrinkled face, large greenish eyes and the characteristic nose leaf of phyllostomid bats is absent. Most interestingly, there is a strong sexual dimorphism: Males not only have more pronounced facial wrinkles than females, but show also a unique fold of skin under the chin that can be raised to cover the lower part of the face like a mask [26]. The distinct sexual dimorphism suggests a use of the face mask in reproduction. However, so far, no information on the reproductive behavior of this elusive species was available.

Here, we report on our recent discovery of an aggregation of *Centurio senex* males in Costa Rica, which provided the first opportunity for behavioral observations on this species in its natural habitat. Goal of this paper is to describe the mating behavior and the basic acoustic repertoire of *C. senex*, thus providing information on one of the most iconic neotropical bat species. Based on the distinct sexual dimorphism we hypothesized that the unique face mask of the *Centurio senex* males is employed during courtship behavior.

# Materials and methods

## Study site

The study was carried out at the Biological Station of the Villa Blanca Hotel (10˚ 12'14N, 84˚ 29'04" W, 1100 masl), located in San Ramón, Alajuela Province, Costa Rica. The predominant habitat is the Very Humid Tropical Forest [27] and the area combines patches of secondary forest with mature continuous forest at an average annual rainfall of ca. 2500 mm per year.

*Centurio senex* males were found along a trail in a small forest patch around an artificial pond. The vegetation combined tall canopy trees with a moderately dense understory layer of small trees. Encountered perches of male *Centurio senex* were marked at night close to the hanging individual with reflective tape. The next morning, perch height above the ground was measured using a 3 m self-retracting metal tape measure (Stanley). In addition, we identified the family of the plants the bats were perched on. As bat perches were spread only over a relatively small area under a closed canopy and some perches were very close to each other ($\leq$ 1m), an attempted GPS—mapping of the individual perches was rather inaccurate but helpful for assessing total lek area size.

## Nocturnal censuses

Following the first observation of perching bats on September 15[th], 2018, a total of 13 visits were made between September 27[th] and October 31[st], when no more perched animals were present. During each visit, perched individuals were hourly monitored from 18:00 to 24:00 hours. As we observed that perches were used day after day, we marked them on October 3[rd] individually in order to have a reference for the following visual observations and video/sound recordings. From this date onwards, we monitored the number of bats present and recorded the use of the marked perches.

For testing whether the frequency of use of the perches differed among perches we performed a G-test, using data from 8 census nights and 44 of the 53 identified perches, which had at least one observation between October 3[rd] and October 28[th], the last day of sampling before the individuals had left the study site.

For determining whether the frequency of use of the perches depended on height above ground we used a Spearman correlation. In order to check for a temporary pattern regarding the use of specific perches over time, we calculated a linear regression model, in which sampling date and time were used as predictive variables for the number of perches with individuals encountered in the entire lek.

## Video and ultrasound recording

We recorded the behavior of bats, using an infrared-sensitive video camera (Sony FDRAX53/B 4K HD) that was shifted each night between several perches, depending on occupancy of the perch. Whenever possible we recorded synchronously audio sequences at a sampling rate of 500 kHz with an CM16/CMPA microphone and an Ultra Sound Gate 116Hm interface (Avisoft Bioacoustics, Berlin, Germany), connected to a laptop computer. We recorded manually

and continuously, resulting in sequences of separate wav-files of 1 min length each. In order to get only sounds from the perched bat, we placed the microphone as close as possible (ca. 0.5–1 m, depending on the sensitivity of the individual to the interference and on the characteristics of the respective perch) without disturbing the perched bat. In total, we accumulated during eight nights at 14 perches a total of 20 hours, 43 minutes and 51 seconds of video recordings. Additionally, we recorded also echolocation calls from bats flying in the understory of the study area, focusing on bats in the size range of C. *senex* that were visible when walking the trails.

## Audio analysis

For a description of the vocalizations of *Centurio senex* we selected recordings from males from 7 different perches, recorded over 6 days. Sequences for analyses were selected based on completeness and quality of the sound recordings and of the synchronous video recordings. Vocalizations were analyzed using the custom-made program Selena (University of Tübingen, Germany). We identified commonly observed vocalizations and measured basic spectral (peak frequency, start and end frequency at -15 dB) and temporal parameters (pulse interval, duration at– 15 dB), and calculated syllable and element durations. All measurements were performed using an FFT of 512 and an overlap of 91.14%, resulting in a frequency resolution after auto padding and interpolation of 981 Hz and a time resolution of 0.09 ms with a dynamic range of 80 dB (Blackman window). Spectrograms were made with Avisoft-Sas Lab Pro (Avisoft Bioacoustics), using a FFT of 512 and an overlap of 75%, resulting in a frequency resolution of 977 Hz and a time resolution of 0.256 ms.

To test whether echolocation parameters of calls emitted while flying were different from those emitted at the perches, we performed Wilcoxon rank sum tests for non-parametric data. For parametric data with equal variances we used a Two Sample t-test. If variances were not equal, we used Welch Two Sample t-tests.

## Results

### Seasonal and daily phenology

On September 15th, 2018, in the early evening hours, bats were observed for the first time at the study site. The animals were hanging exposed and rather calm on small branches in the vegetation. Based on the striking facial structures and the unique and clearly visible facial masks they were identified as male *Centurio senex*. The animals were–compared to other phyllostomid bats—rather tolerant to the cautious approach of an observer, and appeared to be relatively reluctant to leave their perches. The exact same spots with extremely little changes (< 5cm) were occupied over different nights. Perching bats were observed in the study area until October 31st, for a total of 46 days. During this period, we were able to collect data during 13 nights.

Perches of males were distributed over a rather small, ellipsoidal area of approximately 0.13 ha. Perches were found on plants belonging to a variety of families: Siparunaceae, Piperaceae, Heliconiaceae, Myrtaceae, Araceae, Acanthaceae, Melastomataceae, Clusiaceae, and Cyatheaceae.

A first count on September 17th revealed the presence of 17 perched males on site. Over the next weeks this number increased to a maximum of 30 perching males during the same night (October 2nd) before starting to decrease during the first week of October (Fig 1A). Following this decline we did not find any more *Centurio senex* males at the site after October 31st, 2018 and assume that this dissolving of the lek marked the end of the mating period. The total number of perches recorded at the site during the entire study period was 56.

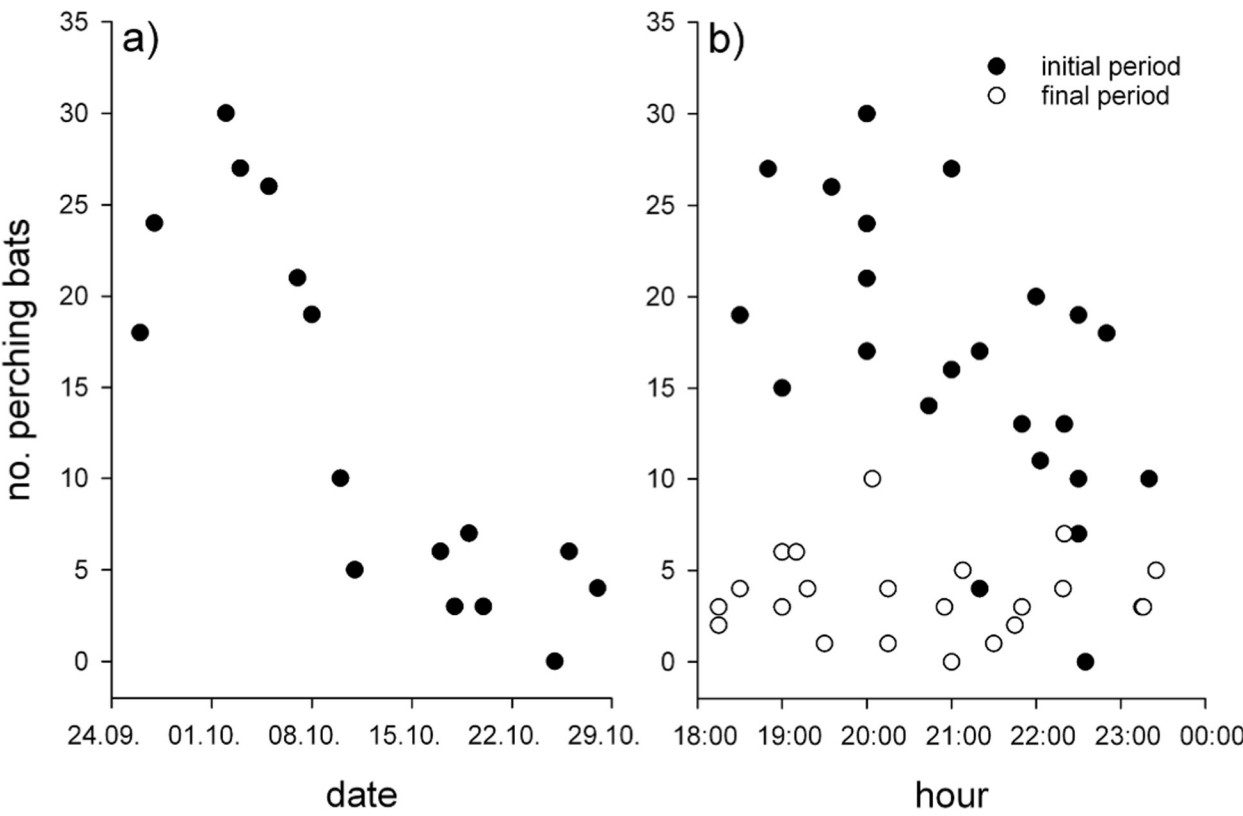

**Fig 1. Phenology of a *Centurio senex* male aggregation.** a) Maximum number of perches occupied by male *C. senex* per night over the entire sampling period. The bats were first observed on September 15th, 2018. b) Development of the number of perching males over the night. Black dots refer to the first half of the study period until October 10th, 2018, white dots refer to the second half.

Some perches were used more frequently than others (G Test; G = 66.34; d.f. = 43; p = 0.013). Especially at the beginning of the monitoring perches were occupied quite regularly, but perch use decreased subsequently. 3 perches were used over the entire monitoring period (S1 Fig). The average height of the perches was 2.84 ±1.26m (mean ±SD; n = 53). Frequency of use observed for a particular perch was not correlated with perch height (Spearman correlation; Rho = -0.24; n = 434; p = 0.11).

No bats were observed at the site during the day, but the animals re-appeared every evening after dusk, around 18:00 hrs. It was common to observe the highest numbers of individuals in the first hours of the night, while after 23:00 hrs we noted more bats in flight and a progressive decrease of the perched individuals (Fig 1B). Around midnight commonly all bats had vacated the perches.

Both, observation date (p <0.001; Fig 1A), and night time (p <0.001; Fig 1B) explain approximately 50% of the variability in the number of perches used throughout study period (F = 23.44; df = 46; p <0.001). In addition, the interaction between these two variables also was significant (p <0.001). In the first half of the study period the number of perching bats decreased markedly towards midnight, while in the second half the (much fewer) bats showed no obvious trend (Fig 1B). Bats were also present at perches during nights with moderate rain, but were rather inactive.

## Vocalizations

*Centurio senex* showed a limited number of stereotypic behavior types that were accompanied by distinct vocalizations. In the following, we describe first these vocalizations before

**Table 1. Analysis of vocalizations of free-flying and perching bats.**

| | PF [kHz] | | SF [kHz] | | EF [kHz] | | BW [kHz] | | PD [ms] | | PI [ms] | | Rep. rate | |
|---|---|---|---|---|---|---|---|---|---|---|---|---|---|---|
| **FREE-FLYING BATS** | | | | | | | | | | | | | | |
| **Echolocation calls** | 88.6 (4) | ± 1.77 | 112.7 (4) | ±3.04 | 76.1 (4) | ±1.35 | 36.6 (4) | ± 4.26 | 1.3 (4) | ± 0.19 | 54.2 (4) | ±19.79 | 23.1 (4) | ±9.02 |
| **PERCHING BATS** | | | | | | | | | | | | | | |
| **Echolocation calls** | 91.2 (7) | ± 4.59 | 116.5 (7) | ± 12.17 | 76.4 (7) | ± 3.27 | 40.1 (7) | ± 9.61 | 1.3 (7) | ± 0.42 | 52.0 (7) | ±12.74 | 26.2 (7) | ± 2.81 |
| **Trills** | 82.9 (7) | ± 6.67 | 102.3 (6) | ± 3.50 | 104.0 (6) | ± 4.06 | 24.5 (6) | ± 9.62 | 1.7 (6) | ± 0.21 | 4.0 (7) | ± 0.28 | 257.1 (7) | ± 9.65 |
| **Wing beats[1]** | - | | - | | - | | - | | - | | 85.5 (7) | ± 6.51 | 14.2 (7) | ± 6.98 |
| **Wing beat calls** | 81.7 (7) | ± 3.05 | 97.9 (5) | ± 5.44 | 64.0 (5) | ± 3.46 | 33.9 (5) | ± 2.97 | 2.4 (5) | ± 0.11 | 86.0 (7) | ± 5.96 | 12.3 (7) | ± 1.99 |
| **High frequency calls** | 159.5 (7) | ± 4.59 | 159.2 (7) | ± 3.39 | 165.5 (7) | ± 4.14 | 6.9 (7) | ± 4.97 | 6.9 (7) | ± 0.69 | - | | - | |
| **Downward modulated calls** | 88.2 (7) | ± 1.96 | 103.4 (7) | ± 4.90 | 90.5 (7) | ± 1.76 | 13.2 (7) | ± 4.38 | 28.9 (7) | ± 4.97 | - | | - | |
| **Whistles[2]** | 28.36 (7) | ± 5.90 | - | | - | | - | | - | | - | | - | |
| **End sweeps** | 10.1 (7) | ± 2.24 | 16.7 (7) | ± 1.39 | 5.1 (7) | ± 0.69 | 11.6 (7) | ± 1.26 | 10.6 (7) | ± 1.46 | - | | - | |

Mean values ± SD of peak frequency (PF), start frequency (SF), end frequency (EF) and bandwidth (WB) in kilo hertz [kHz]; pulse duration (PD) and pulse interval (PI) in milli seconds [ms]; repetition rate (Rep. rate) in calls per second [calls $^*$ s $^{-1}$]; (n) number of sequences analyzed for each parameter. [1]Wing beats are not vocalizations with distinct measurable sound parameters, but noise pulses generated by the bat beating the wings. [2]Reliable measurements of whistle start, end frequency and bandwidth are not possible, because of the inevitably overloaded recordings of this element.

presenting the complete behavioral repertoire composed of the observed movements and the associated acoustic emissions.

**Echolocation behavior.** On all days we were able to record echolocation call sequences of flying *Centurio senex*. When flying along the small trails of the study site the bats emitted very short (1.3ms +/- 0.19 SD) FM echolocation calls with a mean peak frequency of 88.6 kHz +/- 1.77 SD (1st harmonic) and a mean bandwidth of 36.6 kHz +/- 4.26 SD (45 calls from 4 passes) (Table 1). Occasionally, echolocation calls contained during the onset a short, low-intensity QCF- component. While we recorded along the trails in the understory also other, similar-sized phyllostomids, vocalizations of *Centurio senex* differed distinctly, due to the lack of harmonics, both in echolocation calls, as well as in the other vocalizations of the bats: Our recordings of echolocation calls showed consistently only a prominent 1st harmonic, and only extremely rarely we observed a very weak 2nd harmonic.

**Social vocalizations.** Bats were vocalizing spontaneously while perching alone but reacted also to the approach of another individual. A typical audio sequence contained 4 different elements, that were emitted in a distinct order (Figs 2 and 3). The sequence proceeded mostly in the order indicated below, however, elements could also be repeated and the sequence could be interrupted at any point. Courtship songs generally started with an echolocation sequence (ES), followed by a trill call (TC). The approach of another bat often triggered a wing beat sequence (WBS), accompanied by synchronized echolocation calls with reduced bandwidth. Finally, we recorded a complex whistle sequence (WS), consisting of several calls in the high ultrasound range followed by a lower, extremely loud whistling component and an even lower, for humans audible low frequency end sweep (S1 and S2 Audios).

Within these four main elements we identified 8 different syllables (Fig 4, Table 1):

1. Few to many echolocation calls (EC) were contained within echolocation sequences. Echolocation calls of perched bats did not differ significantly from those emitted by flying animals (S1 Table). Echolocation calls were 1.3 ms long, steeply frequency modulated with a peak frequency around 91 kHz, and a high bandwidth of 40 kHz (Figs 3A and 4).

2. Trill syllables (TS) consisted of a down sweep / upsweep connected to the next TS syllable, thus forming a continuous, sinusoid shaped call. Peak frequency was consistently located at

the minimum frequency of these elements, while the maximum frequency of each syllable was very much quieter and for that reason very often not visible in the records. Entire trill sequence contained in general more than 20 TS, were between 80 ms and 130 ms long and were characterized by decreasing peak frequencies of the elements from ca. 120 kHz to 70 kHz (Figs 3B and 4, S2 Table).

3. Wing beat sequences differed from all other sounds produced by *Centurio* in so far as one part is not a vocalization, but a low intensity, yet distinct noise pulse (WB) ranging up to more than 50 kHz that probably was produced by the bat beating its wings at a frequency of ca 14 beats per second (mean number of wing beats in WBS: 7; ±1.0 SD; n = 7, S2 Table). Additionally, each wing beat was ended with a wing beat call (WBC), that seemed to be a lower bandwidth version of an echolocation call EC (Figs 3C and 4).

4. The final vocalization contained a highly stereotypic sequence of four distinct elements. First, an extremely high, 7 ms short call (HF) with a peak frequency of 160 kHz, followed by a longer (29 ms) downward modulated call (DM) descending from 104 kHz to 90 kHz. The following whistle call (WC), a ca. 20 ms long call at ca. 28 kHz was by far the loudest part of the *Centurio senex* call repertoire. The high amplitude of this syllable caused in many recordings a distinct echo reverberating from the surroundings. Last followed an end sweep (E) below 10 kHz that is clearly audible to humans. The extreme dynamic differences among the syllables in this final sequence made it basically impossible to obtain good quality recordings of both the low intensity high-frequency syllables (HF, DM) and the extremely loud low-frequency whistle call (WC) of a sequence, so recordings of the latter were mostly overloaded, with the sonograms showing false harmonic artefacts (Figs 3D and 4).

Within the song elements, syllables could be repeated at sometimes quite variable number. Particularly the echolocation and trill sequences differed both within and among recordings in number of syllables, resulting in different durations of song elements (S2 Table).

## Behavioral observations

**Display behavior.**   From sunset to ca. midnight males spent most of the time on the perches and consistently showed a rather stereotypic behavior. We did not observe anything resembling physical aggression between bats at the perches. Most of the time bats perched with raised skin mask, and lowered it only occasionally. For both raising and lowering of the mask the bats used their thumbs (Fig 5A and 5B).

Phase 1) Over long time periods perched males were just rubbing the wing tips subtly against each other (85.9 ± 8.6% of total observation time), while often moving their head and ears, apparently in a quiet state of alert (S1 Video). Bats emitted in this situation occasionally echolocation sequences (ES), but also the conspicuous descending trill calls (TC).

Phase 2) When another bat approached the perching bat its general alertness increased, indicated by more pronounced ear movements and by directing the whole body towards the visitor. As the visiting bat approached closely, sometimes in a short hovering flight, the perched male began a wing beat sequence (WBS), with each wing beat (WB) followed by a wing beat call (WBC). The encounter escalated with the perched male making a quick, thrusting motion with the entire body towards the visitor and, in most cases (77.9%, n = 181), emitting the very loud and to humans partially audible whistle sequence (WS), whereupon the visitor left (S2 Video). This thrust was accompanied by the perching bat diving deeper into the facial mask, so that both ears were pointing downwards, covering the eyes, and thus

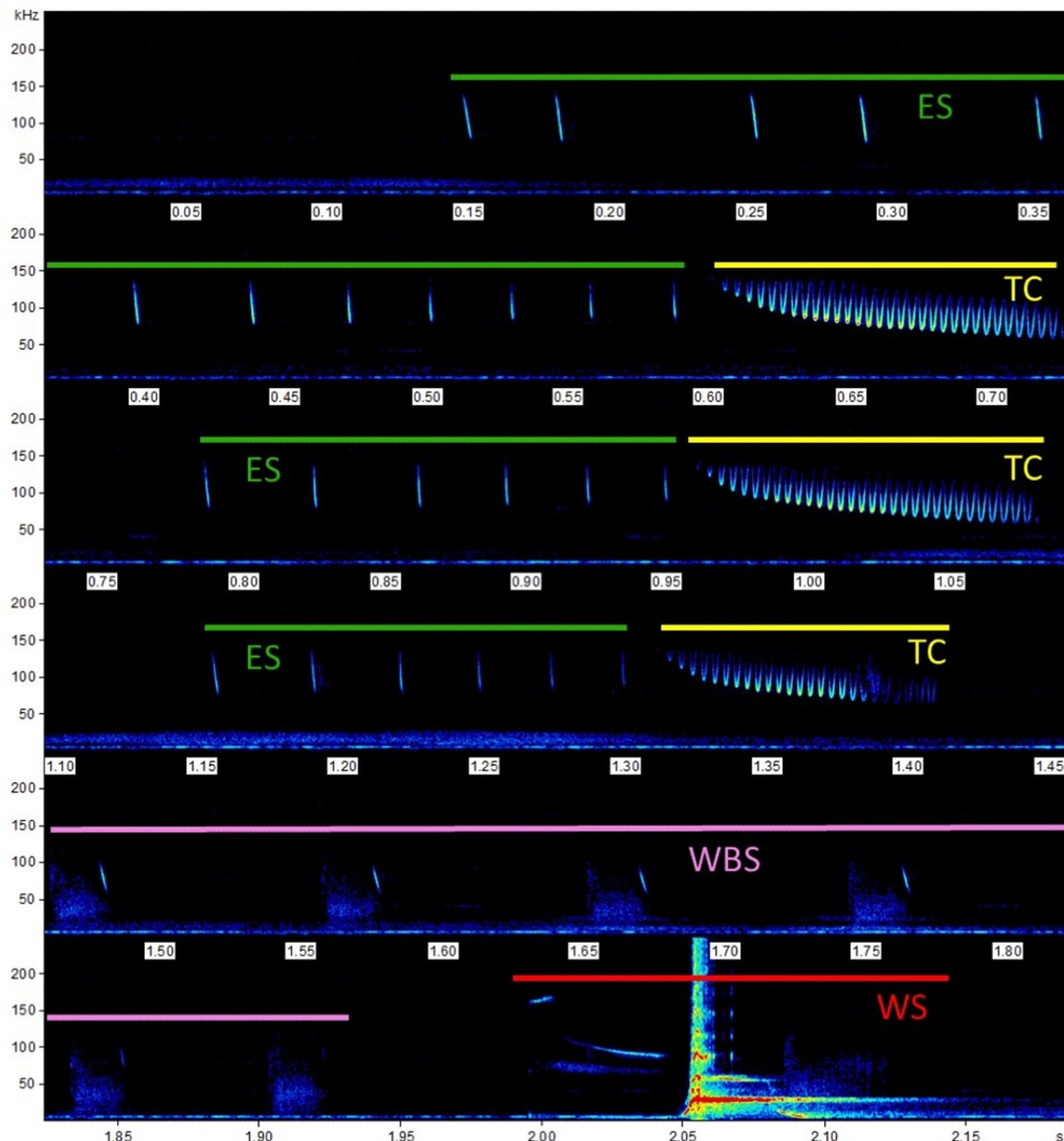

**Fig 2. Example of a typical courtship song of *Centurio senex*.** The four major elements are indicated by color bars: echolocation sequences (ES, green bars), trill calls (TC, yellow bars), wing beat sequences (WBS, pink bar), whistle sequence (WS, red bar. Note harmonic artefacts caused by the inevitably overloaded recording of this always relatively loud element.). The repetition of echolocation and trill sequences resulted in this example in an 8 element song. (Parameters for all spectrograms: Blackman Window, FFT 512, frame size 100%, overlap 75%, resulting in a spectral resolution of 977 Hz and a temporal resolution of 0.256 ms).

minimizing the facial area exposed to the approaching bat (Fig 5C and 5D). The entire phase–from a visitor approaching the perched male to leaving following a whistle call–was in general very short and lasted less than one second. Following the departure of the visitor, the perched male quickly returned to Phase 1) behavior. Unfortunately, the quality of the video did not allow to distinguish reliably between males and females in the visitors. The fast-moving visitors were inevitably blurred; additionally we had to focus on the perched individual, so the

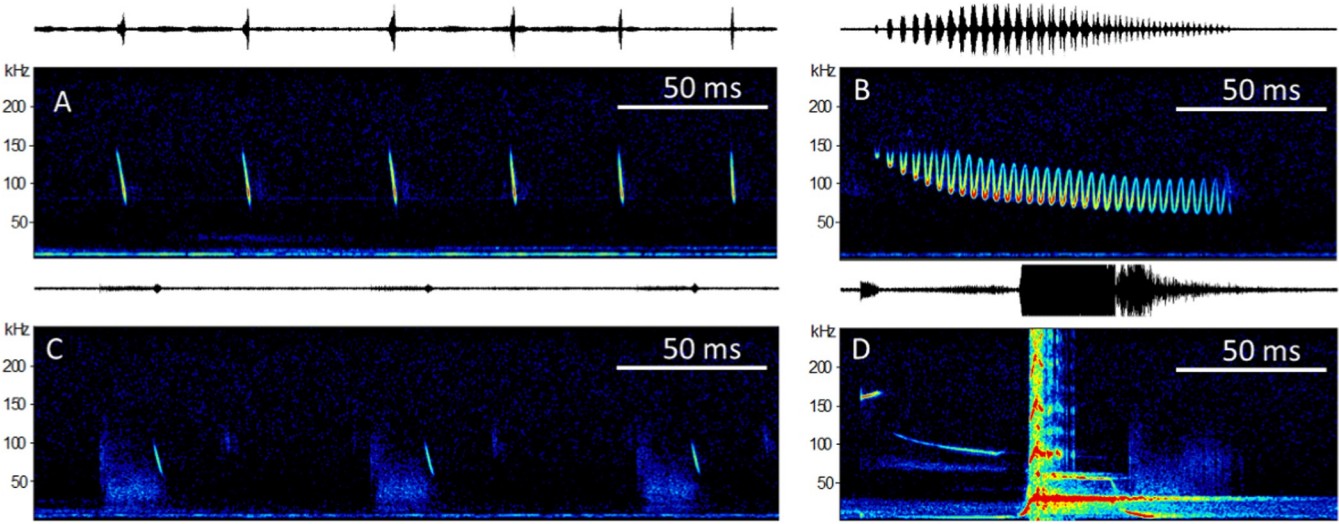

**Fig 3. Song elements.** A complete courtship song is composed of at least four elements: A) echolocation sequence, B) trill call, C) wing beat sequence, D) whistle sequence (Note harmonic artefacts caused by the overloaded recording of this relatively loud element).

hovering individuals were mostly out of focus and it was only rarely possible to see whether a folded face mask was present or not.

In addition to the rather stereotypic response of perching males to visits, we observed the animals occasionally also removing smaller leaves in the immediate surroundings of their perch, and others rubbing their body against branches (n = 10). During the entire recording period, we also observed males on 9 perches grooming themselves 43 times.

## Copulation

On October 10<sup>th</sup> at 19:40 hrs, we observed an approach to a perching male that ended with the usual whistle sequence and the departure of the visitor. Just five seconds later the perched bat was visited again and this time the visitor landed immediately, very close and almost on the back of the perching male. The visitor was a female, clearly recognizable by the lack of a face mask. The male immediately lowered his mask and quickly placed himself in a face-to-face position with the female. Five seconds later he changed his position, moved around the female

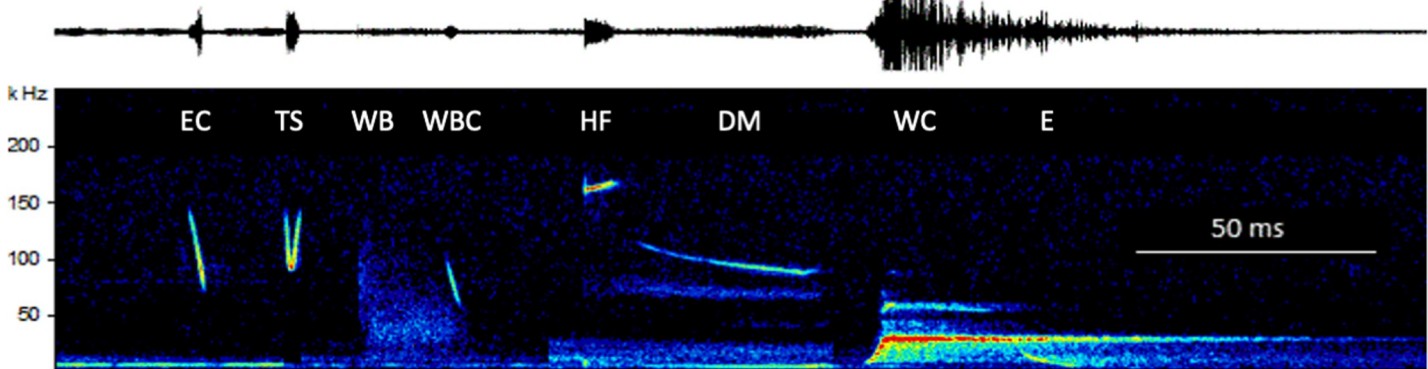

**Fig 4. Syllables in the vocal repertoire of *Centurio senex*:** Echolocation Call (EC), trill (TS), Wing Beat (WB) and associated Wing Beat Call (WBC), high frequency call (HF), downward modulated call (DM), Whistle Call (WC) and an end sweep (E).

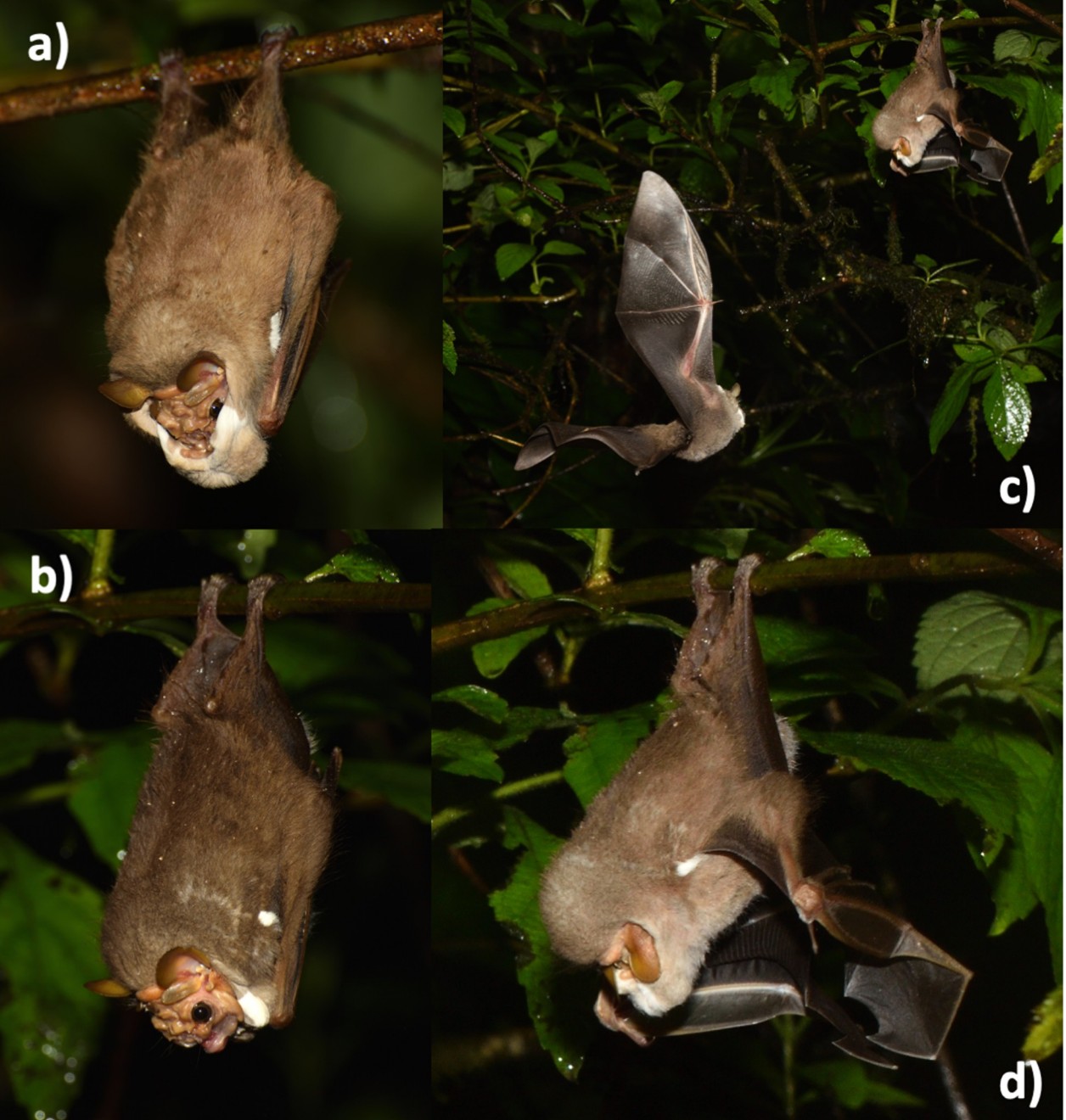

**Fig 5.** Field photographs a) *Centurio senex* male with raised facial mask; b) male with lowered facial mask; c) perching male being approached by another individual; d) detail of same perching male during the approach, just before emitting a whistle sequence.

and pressed his face onto the middle of her back, in a position typical for mating in bats. The visibly vibrating male held the female, using thumb claws and wings, and seemed also to bite into the fur on the lower back of the female, presumably while penetrating. Both bats remained in this position for ca. 30 seconds, and then the female got restless and departed. The male immediately raised the mask again and returned to phase 1) behavior (S3 Video).

In the minute prior to the observed copulation the perching male had been singing several times. A first courtship song (t = - 10 sec before the landing of the female) consisted of an echolocation sequence followed by a trill. The second song (t = - 5 sec) was a complete song with an echolocation sequence, trill, wing beat, and whistle. At the end of the second song a bat approached and got so close to the perching male that the recording shows its echolocation signals during the wing beat sequence, before leaving. The third song (t = -1 sec) was an incomplete song with an echolocation sequence, trill and wing beat sequence, which stopped at the landing of the female (t = 0). During most of the following copulation we recorded no vocalizations, however, close to the end of the act, when the female started to get visibly restless (t = + 33 sec), the bat resumed to performing rather enthusiastically several trilling sequences and continued to do so until the female had left (t = + 38 sec) (S3–S5 Audios).

## Discussion

We observed an accumulation of *Centurion senex* males over a period of several weeks until the animals disappeared. As such an event had never been observed before we refrained from all manipulations of the animals in order not to produce a disturbance that might cause them to abandon the area, which clearly set distinct limitations to our study. Nevertheless, our study provides the first information on echolocation and courtship behavior of a rare and highly interesting frugivorous bat.

### Echolocation

In general, echolocation of phyllostomid bats is characterized by short, frequency-modulated (FM) calls with distinct harmonics [28]. These calls are generally seen as an adaptation to the orientation by echolocation within cluttered space [29], as short and broadband echolocation calls allow a precise spatial resolution [30]. In the temporal domain, the echolocation calls of *C. senex* match this general pattern, however, the frequency domain shows distinct deviations from the family. While the calls of most phyllostomids show highest energy in the 3rd harmonic [28], our recordings from *Centurio senex* showed consistently just the single 1st harmonic and only rarely we observed an extremely weak 2nd harmonic. Emission of echolocation calls with the main energy in the first harmonic is extremely rare in the family and has been observed only in few species, e.g., in the nectarivorous *Phyllonycteris poeyi* [31] or *Leptonycteris yerbabuenae* (Gonzalez-Terrazas, pers. comm.). The single 1st harmonic of *Centurio senex* covers a relatively high bandwidth compared to similar-sized phyllostomids [32, 33], so functionally the species should have access to the same echo information as species covering a larger frequency range by using several harmonics. Harmonics are a common phenomenon in vocalizations, and their absence can be explained by filtering characteristics, e.g., of the vocal tract [34]. Due to its ability to consume hard seeds [25], *Centurio senex* has the relatively shortest skull of all phyllostomid bats and so the vocal tract seems not to be overly long and distinctly limiting the emission of harmonics. However, the relation between skull morphology and sound production in bats is complex and clearly needs more research [35]. The lack of harmonics is also not explained by the facial mask of the perched males, as the echolocation calls of masked perched bats did not differ significantly from those of flying individuals, when *Centurio senex* males usually did not wear the mask.

### Lek mating

Our observations suggest that *Centurio senex* employs a lek mating strategy, as our observations match many characteristics of a "classic" lek [14, 22]: (1) Males aggregate on a relatively small area and show a unique displaying behavior. With their masks raised, male *Centurio*

move over long time periods just their wing tips and vocalize. (2) In the interaction with males, females apparently do not gain more than genetic material. Males do not offer the females access to a particular and limited resource. (3) Although we do not have direct observational data, it is highly unlikely that *Centurio* males contribute to parental care, as this is unknown from the majority of bat species. The sex ratio of *Centurio* is reported to be biased in favor of females (2:1) [36]. (4) Females are free to select and fly to any of the perching males. In addition, *C. senex* shows also a distinct sexual dimorphism, internal fertilization, and high mobility of females that allows them easily to choose between males. We observed the animals until the aggregation of males dissolved. As we were very careful not to disturb the perched animals, we believe that the disappearance of the animals reflected the end of the natural mating period.

Lek courtship behavior seems to be rare in bats, with only few species showing true leks or lek-like behavior [20]. So far, the hammer-headed bat, *Hypsignathus monstrosus* (Pteropodidae) is the only bat reported to form classic leks [10]. Here, males congregate for calling in a specific area, and males use every night the same spot on a branch for calling. The visit of a female to a chosen male is immediately followed by copulation, lasting for 30–60 seconds. Female selection seems to be rather strong, with only 6% of the males obtaining 79% of the observed copulations. Seven more bat species show a lek-like behavior, that either does not fully correspond to the classic lek definition or remains to be confirmed as such, due to the lack of behavioral data [22].

Clearly, more information on *Centurio senex* would be desirable, but based on the available data its behavior resembles the classic lek mating pattern of *Hypsignathus monstrosus*, with calling males relatively close to each other within a relatively small area and females visiting perched males for copulating. Open questions are, e.g., the relative density of males and females in the area, and whether the visitors to the perching males are all females. Also, we recorded just a single copulation, but as we had only a single camera to shift between the perches, we cannot say whether this low number reflects actually a low mating success of the males, or is just the result of our limited sampling.

## Male display behavior

What are the cues that attract females and can be the base for female mate choice and sexual selection in *Centurio senex*? We suspect that the male display behavior might utilize several sensory canals, some effective at distance and others only at close range.

**Acoustics.** Acoustic signals seem to be important since these are presented throughout the time and may advertise the presence of a male also over larger distances. Conspicuous elements are the trilling calls emitted spontaneously during phase 1), which is characterized by the bat moving subtly the fingers against each other, even without another individual close by. Once the perched male is approached by a visitor in phase 2), the acoustic behavior intensifies. The agitated bat starts to beat its wings, which produces broadband noise pulses, and in most cases, the encounter ends with the extremely loud whistle call, upon which the visitor leaves. The impressive volume of this last element—that seems to be rather excessive as it is only emitted once the visitor is very close—might provide information on the physical condition of the perch owner. While the comparably quiet and high frequency trilling calls probably can be heard only over a rather limited range, the final whistle call is not only extremely loud but contains also predominantly lower frequencies, thus it suffers much lower levels of atmospheric attenuation and therefore may be perceived over a larger distance [30]. A close-range function of the trilling calls is also supported by their continued use during the observed copulation. Audio recordings from a single perched male showed rarely trilling calls from other males,

while often there were whistle calls at various levels of intensity, indicating the presence of other males at different distances (see also S2 Video).

The acoustic channel is rather commonly used in bat courtship [e.g., 10, 37–39]. The acoustic repertoire of *Centurio senex* is relatively limited, especially compared with particularly vocal species, such as the emballonurid *Saccopteryx bilineata* [40, 41]. Nevertheless, male quality or even identity could be coded in call intensity as well as in call rate, i.e. the repetition of certain syllables, e.g., in the duration of trill sequences [42]. For future investigations on *Centurio senex*, it might be interesting to measure absolute sound pressure levels, especially of the loud whistle calls and relate calling behavior to body size parameters of the advertising males and their position within the lek site.

Interestingly, although the behavior of *Centurio* males involves distinct vocalization both in the display phase and during the encounters, vocalizations appear to be emitted while the skin mask totally covers the mouth, which could result in a slightly reduced emission volume and a modified frequency spectrum. However, nostrils are mostly uncovered by the raised mask, suggesting at least a partially nasal emission of the observed social vocalizations. Perhaps the unique facial wrinkles which are in males more pronounced than in females, might assist in directing the sonar beam of these calls.

The audible wing beats, representing noise pulses generated through friction, are audible at least on close range. Similar non-vocal sound-production through wing movements has been observed in Old World Fruit bats, although not in a social context but in a rudimentary form of echolocation [43]. In *C. senex*, this behavior might be connected to another morphological peculiarity: Wings of the bat show a distinct pattern of parallel folds between the 5th and the 4th digit [23] that might support the observed sound production. Questions remain, however, as this character is also present in females and seems to not be sexually dimorphic (pers. obs, BRH). Non-vocal sound production appears to be very unusual in bats.

The combined acoustic activity of all males within the relatively small lek area, especially of the whistle calls (S2 Video), and to a lower degree also the trilling calls, might create an acoustic beacon that guides females into the display area [44]. In this context perch position within the lek might be essential, as an orientation of females towards a high acoustic activity would lead them to the center of the lek, resulting in increased mating chances for males occupying center positions. For several bird species, position of the perch has been documented to be a decisive factor [45]. Our results show that not all perches were used equally, with some being occupied over longer periods while others were occupied only occasionally.

**Vision.**   While echolocation is clearly the predominant sensory channel for the nocturnal bats, vision may be also used for obtaining information about the surroundings [46, 47]. Within the phyllostomid family, eye size is particularly variable, indicating species-specific differences in the use of visual information (e.g., [48]). The conspicuously white color of *Centurio senex* males´ facial mask and the white spot near the ears might be visual signals that could improve the visibility. The white color provides a distinct contrast to the darker background vegetation. As the males were displaying at the lek site over 46 days, the perching bats were present during all phases of the moon, including two full moons (25.9.2018, 24.10.2018), so at least on some nights the light available in the understory might be sufficient, at least on close distance. As *Centurio senex* males and females have relatively large eyes (Fig 5B), vision seems to be an important sense for the species, perhaps for landmark orientation during commuting flight, but on close distance in the understory, this could also support mate choice. The extension and shade of the white color of the mask might perhaps even transmit visual information on male quality to interested female vistors. In this context, the observed perch cleaning behavior could improve visibility of a perching male to passing females.

**Olfaction.** Although our strictly observational approach did not allow to collect olfactory data, chemical cues could also be essential elements in male display behavior. Olfaction is used by bats for various purposes, ranging from foraging to social interactions [e.g., 49–51]. The use of olfactory cues by *Centurio senex* is supported by our observation of males rubbing their bodies on the branch, which could represent a scent marking behavior. Early on was reported that male *Centurio senex* have a strong odor in the chin area [26]. Our photos of the naked facial skin indicate a moist surface, suggesting secretion of fluids in the area of the skin pouch. The pouch might even offer a possibility for controlling the release of olfactory signals: As long as the mask is raised there is only limited exposure of the facial skin to the air, while a lowered mask could allow free dissipation of locally produced olfactory volatiles. A similar odor-releasing system has been described from the Greater sac-winged bat, *Saccopteryx bilineata* (Emballonuridae), where males prepare a cocktail of body fluids of different origin in their wing sacs and fan the resulting odour during hovering displays towards the perched female [50]. Images from our study in comparison to males observed on other occasions give the impression that the skin pouch is slightly inflated. It would be interesting to compare the morphological and physical conditions of the skin mask between a displaying male captured from its perch to that of a male captured at another time of the year in a non-reproductive context. Seasonal changes in sexually dimorphic structures, other than in the primary reproductive organs, are uncommon in bats, but are reported for a close relative of *Centurio senex. Pygoderma bilabiatum* males show sexually dimorphic forelimb swellings that seem to vary over the reproductive cycle [52]. Further observations on *C. senex* should, therefore, combine a histological study of the skin mask with the collection of scent-samples from reproductive and non-reproductive males and females.

## Facial mask

Most of the time males perched with raised face mask, even when singing. While this confirms our initial hypothesis, the actual function of the mask for *Centurio senex* males remains still rather unclear. A possible role might be indicated by our single visual record of an approaching visitor, when the perched bat's mask is fully raised (Fig 5D) and exposure of the face is minimized, suggesting a protective function, perhaps for the large eyes, although we never observed any obvious aggressive interactions. Slow motion video recordings would be highly useful to see whether such a protective behavior is a common element of all encounters. An important question in this context would be also whether all visitors are females or whether perching males receive also visits from other males, that are recognizable by their mask. Our observations on the lek courtship behavior of *Centurio senex* males might also have some implications for related species: Besides *Centurio senex* there are also several other species within the subtribe Stenodermatina (short-faced or white-shouldered bats, [53]) that show a similar sexual dimorphism. Besides the already mentioned *Pygoderma* males with its temporal glandular swellings on the wings [52], there is also *Sphaeronycteris toxophyllum*, where males also have a retractable face mask and additionally develop with sexual maturity a permanent, visor-like structure on the forehead. Perhaps also this structure has a potential protective function during a similar courtship behavior as here described for *Centurio senex*. Clearly, we are still far from a full understanding of these unique sexual dimorphic structures.

## Costs of lek mating

Lek mating involves a significant investment for *Centurio senex* males. As perches were occupied between nightfall and ca. midnight, males spent at least 50% of their potential foraging time at the lek site without eating. Most of this time is spent motionless or with subtle finger

movements, complemented by singing and the occasional behavioral and acoustic reactions to a visitor. Besides the cost for the vocalizations [54], additional energy is invested in perch maintenance behavior, such as removing smaller leaves and in the presumed scent marking of the perch. Modification of plants into day roosts is a potentially costly common behavior employed by male and female stenodermatine tent-making bat species [55], however, there are no reports on simple modifications of night roosts such as the small branches used by *C. senex*. Incidentally, the use of an over several days stable and thus predictable perch, while persistently vocalizing might also increase distinctly the predation risk for *Centurio senex* males. This stationary and acoustically conspicuous behavior has the potential to make them a target for acoustically in the rain forest understory hunting predators such as various species of owls.

## Conclusion

Our study summarizes the first behavioral data from a lek of *Centurio senex* males, observed over a period of almost 6 weeks at a Costa Rican highland forest site. The exclusively male bats used stable perches and showed distinct behavioral patterns including an acoustic signaling with stereotypical elements that might serve to attract females and perhaps also to signal male quality. Perching animals were regularly approached by other individuals and in one case such an encounter was followed by a copulation. Males often spent several hours on the perch without leaving and abandoned it only after midnight. This endurance left males with only half of the night time for foraging, so the time spent on the perch is a significant energetic investment and could be a honest signal for male quality. Our study has some obvious shortcomings. While we were extremely lucky to obtain the first observations on the behavior of this interesting species we deliberately refrained from mist-netting bats in order not to scare the animals away from our study site. The downside of this cautious approach is, however, that we still lack essential data, such as the sex ratio at the site, the body condition of individual males or the identity of all visitors. In this context, it would also be highly interesting to assess potential olfactory signals of the perching males, perhaps in the facial region. Hopefully, the encounter of a future lek of *Centurio senex* will allow us to close some of the current gaps in the knowledge on the behavior of one of the most iconic bats of the Neotropics.

## Supporting information

**S1 Fig. Individual perch use between October 4th and October 20th.** Dots indicate the presence of a *Centurio senex* male at the respective perch. Perches occupied during successive surveys are connected by a line.
(TIFF)

**S1 Table. Echolocation parameters showed no significant differences between calls emitted in flight and those from perched animals.**
(DOCX)

**S2 Table. Song elements: Duration and number of syllables recorded at different perches.** (Wing beat sequence: each syllable comprises a wing beat and its associated wing beat call, if present). n is the number of elements analyzed for each song.
(DOCX)

**S1 Video. Display behavior of *Centurio senex*, showing a male with raised mask initially rubbing the wing tips against each other.** At t = 34 sec another animal approaches and the perched male produces an audible whistle call. After the encounter the male returns to wing tip rubbing.
(MOV)

**S2 Video. Interaction between a perching male and a visitor (t = 5–11 sec).** Besides the main interaction, many whistle calls from other males in the lek at widely varying intensity can be heard (e.g., at t = 3, 4.5, 7.5, 9.5, 11.5, 15, 16, 17, 23, 23.5, 24, 26, 34, 36, 36.5, 37.5, 40.5 sec, etc.).
(MOV)

**S3 Video. Copulation between perching male and a visiting female.** After the approach of a hovering individual (t = 15 sec) a female lands (t = 20 sec) and a copulation follows (t = 25–58 sec).
(MOV)

**S1 Audio. Audio file courtship song sequence from Fig 2.** This is the original file with a sampling rate of 500 kHz, that can be played with specialized sound analysis software. Note the clearly audible whistle call at the end of the recording.
(WAV)

**S2 Audio. Same as S1 Audio, just slowed down by a factor 10, so it can be played by standard audio device.**
(WAV)

**S3 Audio. Audio sequence from copulation event.** The copulation started approximately at $t_0$ = 11.25 sec in the original audio file. Note first ($t_0-10$ sec) and second ($t_0-5$ sec) courtship song sequences. The silent phase after $t_0$ covers the actual copulation. Once the female got restless ($t_0 + 33$ sec) the male performed a large number of inspired trilling sounds until the female left ($t_0 + 38$). This is the original file with a sampling rate of 500 kHz, that can be played with specialized sound analysis software.
(WAV)

**S4 Audio. Same as S3 Audio, just slowed down by a factor 10, so it can be played by standard audio device.**
(WAV)

**S5 Audio.**
(WAV)

**S1 Data. Acoustic original data.**
(XLSX)

## Acknowledgments

We appreciate the generous logistical support of the Cloud Forest and Natural Reserve Hotel Villa Blanca. Vivian Rodríguez, Francini Guido, Melissa Rodríguez and Jorge González volunteered for field work. Additionally, we want to thank M. Schöner and an anonymous referee for numerous helpful suggestions and comments.

## Author Contributions

**Conceptualization:** Bernal Rodríguez-Herrera, Marco Tschapka.

**Data curation:** Bernal Rodríguez-Herrera.

**Formal analysis:** Bernal Rodríguez-Herrera, Ricardo Sánchez-Calderón, Victor Madrigal-Elizondo, Paulina Rodríguez, Daniel Zamora-Mejías, Gloria Gessinger.

**Investigation:** Ricardo Sánchez-Calderón, Victor Madrigal-Elizondo, Paulina Rodríguez, Jairo Villalobos, Esteban Hernández, Gloria Gessinger, Marco Tschapka.

**Methodology:** Bernal Rodríguez-Herrera.

**Project administration:** Bernal Rodríguez-Herrera.

**Visualization:** Gloria Gessinger.

**Writing – original draft:** Bernal Rodríguez-Herrera, Marco Tschapka.

**Writing – review & editing:** Bernal Rodríguez-Herrera, Daniel Zamora-Mejías, Gloria Gessinger.

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
