## [Decision Letter · Decision Letter 0]

12 Aug 2020

PONE-D-20-16747

The masked seducers: Lek courtship behavior in the Wrinkle-faced bat Centurio senex (Phyllostomidae)

PLOS ONE

Dear Dr. Rodriguez-Herrera,

Thank you for submitting your manuscript to PLOS ONE. After careful consideration, we feel that it has merit but does not fully meet PLOS ONE’s publication criteria as it currently stands. Therefore, we invite you to submit a revised version of the manuscript that addresses the points raised during the review process.

Specifically, it would be important to address the reviewers' comments about clarifying some parts of the paper and including appropriate caveats about the results.

We look forward to receiving your revised manuscript.

Kind regards,

Vivek Nityananda

Academic Editor

PLOS ONE

Journal Requirements:

We note that one or more of the authors are employed by a commercial company: Hotel Villablanca.

2.1. Please provide an amended Funding Statement declaring this commercial affiliation, as well as a statement regarding the Role of Funders in your study. If the funding organization did not play a role in the study design, data collection and analysis, decision to publish, or preparation of the manuscript and only provided financial support in the form of authors' salaries and/or research materials, please review your statements relating to the author contributions, and ensure you have specifically and accurately indicated the role(s) that these authors had in your study. You can update author roles in the Author Contributions section of the online submission form.

2.2. Please also provide an updated Competing Interests Statement declaring this commercial affiliation along with any other relevant declarations relating to employment, consultancy, patents, products in development, or marketed products, etc. 

Reviewers' comments:

Reviewer's Responses to Questions

**Comments to the Author**

1. Is the manuscript technically sound, and do the data support the conclusions?

Reviewer #1: Yes

Reviewer #2: Yes

2. Has the statistical analysis been performed appropriately and rigorously? 

Reviewer #1: Yes

Reviewer #2: Yes

3. Have the authors made all data underlying the findings in their manuscript fully available?

Reviewer #1: Yes

Reviewer #2: Yes

4. Is the manuscript presented in an intelligible fashion and written in standard English?

Reviewer #1: Yes

Reviewer #2: Yes

5. Review Comments to the Author

Reviewer #1: The authors described for the very first time Lek mating behavior in the bat Centurio senex, this is a valuable and exciting finding. Furthermore, the manuscript is expertly prepared. It is extremely rare to read an article that so thoroughly anticipates and answers questions/concerns. Both the topic, the evaluation of the data, and the quality of the writing, made this a true pleasure to review.

My only very minor comments are as follows:

- Even if its just observational, as the bats weren't individually marked, do the authors have an idea if the same roosts were used by the same individuals on consecutive days (that is: had some of them maybe some natural markings that made them identifiable)?

- Although all 53 perches were marked, I am assuming not all were fitted with cameras & microphones, thus I would encourage the authors to acknowledge this and discuss the possibility that other copulation events might have been missed. The authors present compelling evidence that this is in fact mating behaviors but with only one recorded event I believe it necessary to address this either lack of success by the males or impossibility to survey all perches simultaneously allowing for some missed events.

- A table or set of graphs that shows the statistical comparison of the features of perched and in flight echolocation would be very useful. Also, the metabolic cost of perched echolocation and potential uses in communication has been discussed for other species (for example: Front Physiol. 2013; doi: 10.3389/fphys.2013.00066. Metabolic costs of bat echolocation in a non-foraging context support a role in communication. D. Dechmann, M. Wikelski, H. van Noordwijk, C. Voigt, S. L. Voigt-Heucke.)

- The authors discuss the differences in the composition of the songs in regards to the number of elements, but, were there differences in the spectro-temporal signatures of the syllables among the bats that may allow for identification of individuals? (Bohn KM, Schmidt-French B, Schwartz C, Smotherman M, Pollak GD (2009) Versatility and Stereotypy of Free-Tailed Bat Songs. PLoS ONE 4(8): e6746. https://doi.org/10.1371/journal.pone.0006746)

- For the behavioral observations it would be useful to have percentages of time, for example the authors describe "94.1 ± 13.89 min of total observation time" which one needs to go back further up to check how long was the total observation time. Adding % would aid the reader to quickly understand if this is a significant amount of time.

- Are all visitors female? The authors mention this question in the discussion do not explain why this is not something that can be know with the collected data. What is the limitation to answering this question with the collected data? viewing angle for the visiting bat? few perch sites being recorded and this identification being hard to do with the naked eye? Any or all of this are valid, it just would be good to explain why this is not known.

- The authors mention that sound production with the wings in bats is rare and this is true, yet they might want to cite a paper that shows a species of fruit bats that use wing beat sounds for sonar (Nonecholocating Fruit Bats Produce Biosonar Clicks with Their Wings. Arjan Boonman, Sara Bumrungsri, Yossi Yovel. https://doi.org/10.1016/j.cub.2014.10.077)

I would also recommend citing the two following review articles:

- This one reviews auditory processing of communication in bats and points out which species until the data it was published have been described to show songs. This is relevant the introduction. (Salles, A., Bohn, K. M., & Moss, C. F. (2019). Auditory communication processing in bats: What we know and where to go. Behavioral Neuroscience, 133(3), 305–319. https://doi.org/10.1037/bne0000308)

- This other one more broadly reviews social communication in bats and includes detailed descriptions of acoustic, visual, tactile and olfactory information different species obtain through these sensory modalities. (Chaverri, G., L. Ancillotto, and D. Russo. 2018. Social communication in bats. Biological Reviews, 93: 1938-1954).

Reviewer #2: Review for: Rodriguez-Herrera et al.: The masked seducers: Lek courtship behavior in the Wrinkle-faced bat Centurio senex (Phyllostomidae)

The authors report on an observational study on the bat species Centurio senex in Costa Rica. The bat is famous for its rather strange looking facial mask. Beyond, not much is known about the species. This is also true for lek courtship behavior in bats.

Overall, I very much enjoyed reading the manuscript. The fact that the study is more or less purely descriptive and does not contain many hypothesis, predictions and experiments is in this case a plus. It allows the authors to tell the readers about their observations free from any bias. Moreover, especially in the discussion the authors address many questions and give ideas for future studies, which can then be conducted based on hypothesis and experiments. The manuscript thus can be seen as an important starting point for further studies on C. senex, lek behaviour in bats and more generally courtship behaviour in animals.

I only have some minor points, which I will address in the following:

Introduction

“Additionally, variables such as time and place where bats will show courtship behavior are for many species extremely difficult or impossible to predict” … and to access, I would say.

Hypothesis: The only hypothesis in the paper refers to the facial mask. However, I am missing some predictions that explain this hypothesis. What might be the function of the mask? In example: Might certain features of the mask (e.g. size) act as a sign for good genes? Might it be a protection against competing males (as the authors suggest in the discussion)?

Materials and Methods

“On October 3rd, perches were individually marked.” Why only on this date? Or do the authors mean: From October 3rd on, perches were individually marked?

Ultrasound recording

Regarding the equipment for the USG 116Hm: Which microphone did the authors use? The authors write that the microphone was placed “as close as possible”? How close was this on average?

Did the authors trigger manually, automatically or did they choose continuous recordings? If the recording was done automatically, which settings did they choose?

“Additionally, we recorded also echolocation calls from bats flying in the study area.” Which bats were recorded? Does this refer to C. senex or any bat species flying in the study area?

Results

Were there any obvious reasons, why the number of perching males started to decrease in October? Could this have been the end of mating season, rainy/dry season or due to monitoring? If so, it should be discussed in the Discussion section.

Average height of the perches: In the M&M section it is missing how these measures were done.

Echolocation behavior: For the bandwidth the SD is missing in the text.

Discussion

Vision: Although this cannot be proven, I would guess that the males’ facial mask itself might be an indication that females select males based on visual information.

Olfaction: This is just an idea for future tests that came to my mind. Might the facial mask have an olfactory function similar to the pouches of Saccopteryx? Secreted fluids would than be protected as long as the mask is lowered. When the mask is raised, pheromones could be spreaded through the air.

Best wishes

Michael Schöner

6. PLOS authors have the option to publish the peer review history of their article (what does this mean?). If published, this will include your full peer review and any attached files.

Reviewer #1: No

Reviewer #2: **Yes: **Michael Schöner

---

## [Author Response · Author response to Decision Letter 0]

27 Sep 2020

Video 3 now is available.

Comments from Dr. Vivek Nityananda Academic Editor PLOS ONE 

Comment 1: Please ensure that your manuscript meets PLOS ONE's style requirements, including those for file naming.

Response: The manuscript “The masked seducers: Lek courtship behavior in the Wrinkle-faced bat Centurio senex (Phyllostomidae)” meets PLOS ONE's style requirements as suggested. 

Comment 2: "The authors have declared that no competing interests exist." 

We note that one or more of the authors are employed by a commercial company: Hotel Villablanca.

Response: Hotel Villa Blanca did not play a role as funding source nor in the study design, data collection and analysis, decision to publish, or preparation of the manuscript, however they provided financial support in the form of salaries for two of the authors, yet not for working on this project. We have reviewed the statements relating to the author contributions, and we made sure to specifically and accurately indicate the role these authors had in the study. We have updated author roles in the Author Contributions section of the online submission form. 

Additionally, we have included the following statement within the amended Funding Statement “The owner of the study site, Hotel Villa Blanca, provided support in the form of salaries for authors J.V. and E.H., but did not have any role in the study design, data collection and analysis, decision to publish, or preparation of the manuscript. The specific roles of these authors are defined in the ‘author contributions’ section. Furthermore, we state and explain this role within the updated Funding Statement. 

We have also provided an updated Competing Interests Statement declaring this commercial affiliation along with any other relevant declarations relating to employment of the mentioned authors. 

Within the Competing Interests Statement, we confirmed that this commercial affiliation did not alter our adherence to all PLOS ONE policies on sharing data and materials. We included the following statement as required by PLOS ONE "This does not alter our adherence to PLOS ONE policies on sharing data and materials.” 

Comment 3: We note that you have indicated that data from this study are available upon request. PLOS only allows data to be available upon request if there are legal or ethical restrictions on sharing data publicly.

Response: In addition to two tables provided as supplementary information we have now uploaded the data set necessary to replicate our key findings as Supporting Information entitled “S4 Original Data The masked seducers”. 

Comments from Reviewer #1

“The authors described for the very first time Lek mating behavior in the bat Centurio senex, this is a valuable and exciting finding. Furthermore, the manuscript is expertly prepared. It is extremely rare to read an article that so thoroughly anticipates and answers questions/concerns. Both the topic, the evaluation of the data, and the quality of the writing, made this a true pleasure to review. 

My only very minor comments are as follows:”

Comment 1: Even if it's just observational, as the bats weren't individually marked, do the authors have an idea if the same roosts were used by the same individuals on consecutive days (that is: had some of them maybe some natural markings that made them identifiable)?

Response: We totally agree with the relevance of being able to identify and track individuals, however, in situ C. senex individuals did unfortunately not show any evident natural marks that permitted an individual identification. In addition, as we were anxious not to lose this unique opportunity, we decided not to perform any marking or capture activities in order not to scare the animals away from the site. We apologize but we have no further evidence or data to add, although we believe that the perches are used consistently by the same individuals. Hopefully next time… 

Comment 2: Although all 53 perches were marked, I am assuming not all were fitted with cameras & microphones, thus I would encourage the authors to acknowledge this and discuss the possibility that other copulation events might have been missed. The authors present compelling evidence that this is in fact mating behaviors but with only one recorded event I believe it necessary to address this either lack of success by the males or impossibility to survey all perches simultaneously allowing for some missed events..

Response: We had only a single camera set up that was shifted between perches and we include this information now in the Methods. We added also a remark in the Discussion on the problems of distinguishing between a “real” low copulation success of males and low observation probability of the event, due to a single available camera for sometimes more than 30 perches. 

Comment 3: A table or set of graphs that shows the statistical comparison of the features of perched and in-flight echolocation would be very useful. 

Response: We added to the Supplementary Information a table (S2 table) containing the statistical comparison between the echolocation call parameters of calls emitted by animals in flight and from animals at perches. 

Also, the metabolic cost of perched echolocation and potential uses in communication has been discussed for other species (for example: Front Physiol. 2013; doi: 10.3389/fphys.2013.00066. Metabolic costs of bat echolocation in a non-foraging context support a role in communication. D. Dechmann, M. Wikelski, H. van Noordwijk, C. Voigt, S. L. Voigt-Heucke.).

Response: We cite now the suggested paper in our paragraph on metabolic costs of courtship behaviour.

Comment 4: The authors discuss the differences in the composition of the songs in regards to the number of elements, but, were there differences in the spectro-temporal signatures of the syllables among the bats that may allow for identification of individuals? (Bohn KM, Schmidt-French B, Schwartz C, Smotherman M, Pollak GD (2009) Versatility and Stereotypy of Free-Tailed Bat Songs. PLoS ONE 4(8): e6746. https://doi.org/10.1371/journal.pone.0006746 .

Response: While we did not observe any obvious differences in spectro-temporal parameters, we can imagine that there might be differences between individuals for some of the measured parameters. However, for really attempting such an investigation it would be mandatory to have a sample of unequivocally marked individuals, preferably with additional body metrics. With the current data set we are not able to distinguish reliably between individual variability or individual signatures. While we therefore believe this approach to be outside the scope of the current manuscript, we certainly hope to conduct such a study in the future and cite the mentioned paper now in the manuscript. 

Comment 5: For the behavioral observations it would be useful to have percentages of time, for example the authors describe "94.1 ± 13.89 min of total observation time" which one needs to go back further up to check how long was the total observation time. Adding % would aid the reader to quickly understand if this is a significant amount of time. 

Response: Thanks for the suggestion. We changed in the Results the values mentioned for the percentage of time spent by bats in phase 1.

Comment 6: Are all visitors female? The authors mention this question in the discussion do not explain why this is not something that can be know with the collected data. What is the limitation to answering this question with the collected data? viewing angle for the visiting

 bat? few perch sites being recorded and this identification being hard to do with the naked eye? Any or all of this are valid, it just would be good to explain why this is not known.

Response: Live observations are not possible in these quickly moving, nocturnal and small animals and unfortunately also the video-recordings do not help very much to obtain the required details. The fast-moving visiting bats are inevitably blurred due to the frame rate / time resolution of the video camera, and additionally we could only focus on the perched individual, so the hovering individuals were mostly out of focus. This results in poor quality that did not allow a reliable assessment of the visiting animals through the absence or presence of the mask. We included a short statement in the Results explaining the problem.

Comment 7: The authors mention that sound production with the wings in bats is rare and this is true, yet they might want to cite a paper that shows a species of fruit bats that use wing beat sounds for sonar (Nonecholocating Fruit Bats Produce Biosonar Clicks with Their Wings. Arjan Boonman, Sara Bumrungsri, Yossi Yovel. https://doi.org/10.1016/j.cub.2014.10.077)

Response: Thanks for the suggestion, we cite this paper now in the Discussion.

Comment 8: I would also recommend citing the two following review articles: 

- This one reviews auditory processing of communication in bats and points out which species until the data it was published have been described to show songs. This is relevant the introduction. (Salles, A., Bohn, K. M., & Moss, C. F. (2019). Auditory communication processing in bats: What we know and where to go. Behavioral Neuroscience, 133(3), 305–319. https://doi.org/10.1037/bne0000308) 

- This other one more broadly reviews social communication in bats and includes detailed descriptions of acoustic, visual, tactile and olfactory information different species obtain through these sensory modalities. (Chaverri, G., L. Ancillotto, and D. Russo. 2018. Social communication in bats. Biological Reviews, 93: 1938-1954).

Response: Both references are now included in the Introduction.

Comments from Reviewer #2 

“Overall, I very much enjoyed reading the manuscript. The fact that the study is more or less purely descriptive and does not contain many hypothesis, predictions and experiments is in this case a plus. It allows the authors to tell the readers about their observations free from any bias. Moreover, especially in the discussion the authors address many questions and give ideas for future studies, which can then be conducted based on hypothesis and experiments. The manuscript thus can be seen as an important starting point for further studies on C. senex, lek behaviour in bats and more generally courtship behaviour in animals. 

I only have some minor points, which I will address in the following:

Comment 1: Introduction 

“Additionally, variables such as time and place where bats will show courtship behavior are for many species extremely difficult or impossible to predict” … and to access, I would say. 

Response: Thank you for the suggestion. We have modified the first sentence as follows “Additionally, variables such as time and place where bats will show courtship behavior are for many species extremely difficult or impossible to predict or access.”

Hypothesis: The only hypothesis in the paper refers to the facial mask. However, I am missing some predictions that explain this hypothesis. What might be the function of the mask? In example: Might certain features of the mask (e.g. size) act as a sign for good genes? Might it be a protection against competing males (as the authors suggest in the discussion)?

Response: Our problem was to set up a hypothesis that is really testable. As our study was totally hands-off and purely observational with no information on individual animals, our options were extremely limited. For that reason, unfortunately, none of the suggested predictions would be really verifiable, as the available data would not allow testing for a correlation of good genes or even just of reproductive success with features of the mask. Similarly, although we really like the idea of the mask serving as a physical protection against competing males, we cannot even reliably distinguish between male and female visitors and the details of the interaction between perched and visiting individuals are rather unclear. With a lot of luck we managed to get a single acceptable photo of a visit (Fig. 5c), but for really obtaining data we would have needed a high-speed camera, which was not possible in the short time available. With all due respect we would therefore prefer to stay with our very simple, yet verifiable hypothesis, that is based on the distinct sexual dimorphism known already before our study, and then discuss, based on our findings and observations, potential functions of the facial mask, including also the protective function, in the Discussion. 

Comment 2: Materials and Methods 

“On October 3rd, perches were individually marked.” Why only on this date? Or do the authors mean: From October 3rd on, perches were individually marked?

Response: To clarify this comment we have rewritten this section in the MM section to make our statement clearer. Also, we describe in the Results this point in detail as follows “The animals were – compared to other phyllostomid bats - rather tolerant to the cautious approach of an observer, and appeared to be relatively reluctant to leave their perches. The exact same spots with extremely little changes (< 5cm) were occupied over different nights. Perching bats were observed in the study area until October 31st, for a total of 46 days. During this period, we were able to collect data during 13 nights.”

Comment 3: Ultrasound recording 

Regarding the equipment for the USG 116Hm: Which microphone did the authors use? The authors write that the microphone was placed “as close as possible”? How close was this on average? Did the authors trigger manually, automatically or did they choose continuous recordings? If the recording was done automatically, which settings did they choose? 

“Additionally, we recorded also echolocation calls from bats flying in the study area.” Which bats were recorded? Does this refer to C. senex or any bat species flying in the study area?

Response: We added the missing information on the microphone, microphone placement and on our recording strategy at the porches and along the trails in the understory. We recorded only phyllostomid bats along the trails, and while the echolocation calls are rather generic within phyllostomids in general, those of Centurio senex are different, based on harmonic structure. We explain this in the Results. 

Comment 4: Results 

Were there any obvious reasons, why the number of perching males started to decrease in October? Could this have been the end of mating season, rainy/dry season or due to monitoring? If so, it should be discussed in the Discussion section.

Response: As we were really careful not to disturb the animals we believe that the disappearance of the animals was due to the end of the mating period. We added a sentence in the Results section to clarify this statement and also in the Discussion. 

Average height of the perches: In the M&M section it is missing how these measures were done. 

Response: We describe now in MM how the average height of perches was measured.

Echolocation behavior: For the bandwidth the SD is missing in the text.

Response: SD of the bandwidth was added.

Comment 5: Discussion 

Vision: Although this cannot be proven, I would guess that the males’ facial mask itself might be an indication that females select males based on visual information. 

Olfaction: This is just an idea for future tests that came to my mind. Might the facial mask have an olfactory function similar to the pouches of Saccopteryx? Secreted fluids would then be protected as long as the mask is lowered. When the mask is raised, pheromones could be spreaded through the air. 

Response: Fascinating idea, thank you! Indeed, this would be very interesting to investigate in the future! Currently, many raising questions cannot be answered. We added the point about the facial mask per se providing optical information to interested females, as well as the possibility of the mask aiding in dispensing olfactory signals in the Discussion.

---

## [Editor Report · Decision Letter 1]

8 Oct 2020

The masked seducers: Lek courtship behavior in the Wrinkle-faced bat Centurio senex (Phyllostomidae)

PONE-D-20-16747R1

Dear Dr. Rodriguez-Herrera,

We’re pleased to inform you that your manuscript has been judged scientifically suitable for publication and will be formally accepted for publication once it meets all outstanding technical requirements.

Kind regards,

Vivek Nityananda

Academic Editor

PLOS ONE
---

## [Editor Report · Acceptance letter]

19 Oct 2020

PONE-D-20-16747R1 

The masked seducers: Lek courtship behavior in the Wrinkle-faced bat *Centurio senex* (Phyllostomidae) 

Dear Dr. Rodriguez-Herrera:

I'm pleased to inform you that your manuscript has been deemed suitable for publication in PLOS ONE. Congratulations! Your manuscript is now with our production department. 

Kind regards, 

on behalf of

Dr. Vivek Nityananda 

Academic Editor

PLOS ONE